# PVDF/Clay Spheres Obtained through Phase Inversion for Cu Ion Removal

**DOI:** 10.3390/polym15122643

**Published:** 2023-06-10

**Authors:** Gabriel C. Dias, Mayk F. Cardoso, Alex O. Sanches, Mirian C. Santos, Luiz F. Malmonge

**Affiliations:** 1School of Natural Sciences and Engineering, São Paulo State University (UNESP), Ilha Solteira 15385-000, SP, Brazil; 2Institute of Chemistry Araraquara, São Paulo State University (UNESP), Araraquara 14800-900, SP, Brazil; 3Alternative Technologies of Detection, Toxicological Evaluation and Removal of Micropollutants and Radioactives (INCT-DATREM), Institute of Chemistry, São Paulo State University (UNESP), Araraquara 14800-900, SP, Brazil

**Keywords:** polymer composite, PVDF, clay spheres, drip method, adsorptive copper

## Abstract

In this study, spheres of poly (vinylidene fluoride)/clay were synthesized using an easy dripping method (also known as phase inversion). The spheres were characterized by scanning electron microscopy, X-ray diffraction, and thermal analysis. Finally, application tests were carried out using commercial cachaça, a popular alcoholic beverage in Brazil. The SEM images revealed that during the solvent exchange process for sphere formation, PVDF tends to form a three-layered structure with a low-porosity intermediate layer. However, the inclusion of clay was observed to reduce this layer and also widen the pores in the surface layer. The results of the batch adsorption tests showed that the composite with 30% clay content in relation to the mass of PVDF was the most effective among those tested, with the removal of 32.4% and 46.8% of the total copper present in the aqueous and ethanolic media, respectively. The adsorption of copper from cachaça in columns containing cut spheres resulted in adsorption indexes above 50% for samples with different copper concentrations. Such removal indices fit the samples within the current Brazilian legislation. Adsorption isotherm tests indicate that the data fit better to the BET model.

## 1. Introduction

Metals are known for their toxic capacity and for accumulating in the environment. Many of these metals play important roles in living organisms and are indispensable nutrients. However, contact with a high concentration of these substances due to human activities is related to a variety of diseases, including cancer. Copper is one of the most industrially useful metals due to its malleability, high electrical conductivity, high thermal conductivity, low corrosion, and alloying abilities. Due to these characteristics, it is widely applied in machine production, construction, transportation, and military weapons production [1,2,3,4,5,6,7,8,9,10]. In Brazil, copper residues above the permitted value appear in consumer products, mainly in alcoholic beverages such as cachaça; this is related to unsophisticated artisanal production and the use of inadequate equipment.

Among the alternatives for heavy metal removal, adsorption has emerged as an interesting process. Over time, several low-cost adsorbents have been developed from industrial or agricultural waste, as well as from modified biopolymers [4,5]. Clays are an interesting alternative, with a good cost–benefit ratio as they are abundantly found. Their high surface area allows for good adsorption of liquids and heavy metals [4,5,6,7,8].

In this context, poly (vinylidene fluoride) (PVDF) is an excellent option due to its excellent chemical resistance and good mechanical properties, in addition to good thermal stability [9,10]. These characteristics of the material make it a candidate for numerous applications, such as microfiltration, ultrafiltration, distillation, gas separation, ion exchange, and pollutant removal.

Therefore, in this study, PVDF spheres incorporated with clay were synthesized and characterized. The resulting material was applied in batch adsorption tests for copper removal, and its effectiveness was evaluated in two different media: water and water/ethanol.

## 2. Materials and Methods

### 2.1. Materials

PVDF (SOLEF 1008/1001, Solvay Fluoropolymers Mw = 244,000 g/mol) in powder form was used as the polymer. N, N-dimethylformamide (DMF—99.8%; LabSynth Products Laboratories Ltd., Sao Paulo, Brazil) was used as a solvent. Montmorillonite K-10 {(Na,Ca)_0.33_(Al,Mg)_2_(Si_4_O_10_)-(OH)_2_.nH_2_O} purchased from Sigma Aldrich (surface area, 250 g/m^2^) was used as the clay. Ethyl alcohol (99.5%) was purchased from the company LabSynth Products Laboratories. The copper solutions were prepared using a standard solution of copper (Cu^2+^) with a concentration of 1000 ppm.

### 2.2. Preparation of PVDF/Clay Mixtures

Initially, the clay was placed in a desiccator for 24 h for moisture removal. Following this, it was dispersed in DMF utilizing a magnetic stirrer at percentages of 5%, 10%, 20%, and 30% m/m PVDF. The mixture was continuously stirred for a period of 12 h. Subsequently, a PVDF/DMF solution was prepared (30% m/V) by dissolving 3 g of PVDF in 10 mL of DMF at a temperature of 70 °C, under constant stirring for a duration of 90 min. This concentration was found to be more effective in suspending the clay for a sufficient period in order to prepare the spheres. The DMF/clay mixture was then added to the PVDF solution, and the resultant mixture was stirred at 70 °C under constant stirring for a duration of 30 min. Thereafter, the mixture was allowed to cool to room temperature while still being stirred. Table 1 presents a detailed account of the quantities of materials utilized per 10 mL of solution and per concentration of the composite.

### 2.3. Preparation of Spheres

Spheres were produced by the drip method. In this technique, a polymeric solution was immersed in a coagulant bath consisting of a nonsolvent. As a result, phase separation occurs as the solvent in the polymeric solution is exchanged with the nonsolvent in the coagulant bath. Contents exceeding 30% of clay hindered the formation of spheres, thus presenting a limiting factor. A syringe was used to drop the solution into a beaker containing distilled water. Upon contact with water, each drop precipitated in the form of a sphere (Figure 1). After all the solution was dropped into the beaker, the spheres remained immersed for 4 h to ensure solvent exchange. Subsequently, the spheres were placed in an incubator to dry for 12 h at 50 °C. The average diameter of the spheres was approximately 2 mm.

### 2.4. Characterizations

Scanning electron microscopy (SEM) measurements were performed in a Zeiss instrument (model EVO LS15, Oberkochen, Germany). The samples were fixed in stubs with double-sided conductive tape, and a thin gold layer was deposited. To view the interior of the spheres, they were fractured using liquid nitrogen. X-ray diffractometry measurements were carried out using a Shimadzu XRD-6000 (Kyoto, Japan) X-ray diffractometer with Cu Kα radiation (λ = 1.54056 Å), scanning at a speed of 1°/min within a 2θ angular range of 5° to 40°. Thermogravimetric (TG) analyses were conducted using a TA Instruments model Q-600. The measurements were performed on an alumina crucible at a heating rate of 10 °C/min. The measurements were carried out in the temperature range of 30 to 1000 °C under a dynamic nitrogen atmosphere (100 mL/min).

The zero-charge potential (PZC) was determined using the potentiometric method. Aqueous and ethanolic (95/5 water/ethanol) solutions (20 mL) with a pH of 1–12 were prepared. Sodium hydroxide (NaOH) was used for basic pH, and hydrochloric acid (HCl) for acid pH. A total of 100 mg of PVDF/clay spheres was added to each solution, and the solutions were stirred constantly for 24 h. The spheres were then removed, and the pH was measured; the obtained value was considered to be the final pH. From the graph of the initial pH versus the final pH (Figure 2), it was possible to determine the PZC of the material.

### 2.5. Adsorption Tests

The copper ion concentration was determined using a high-resolution flame atomic absorption spectrometer with a continuous source (ContrAA 300—Analytik Jena^®^, Jena, Germany) and a 300 W xenon short-arc lamp (XBO 301—GLE^®^, Berlin, Germany).

Batch adsorption tests were performed in deionized water and the water/ethanol medium (95/5). The pH of the adsorption solutions was adjusted to the PZC determined for each medium. In an adsorption process involving a solution, the solid adsorbent may have a pH below the PZC, resulting in a negatively charged surface. This will favor the adsorption of anions. Conversely, when the pH of the solution is above the PZC, the solid will become positively charged, favoring the adsorption of cations.

The effects of clay content on adsorption in both media were investigated. Samples of pure PVDF and PDVF/clay (approximately 200 mg) at a content of 5, 10, 20, and 30% (m/m) were used. The copper concentration of the solution was 5 mg/L, and the same volume, 25 mL, was used in all the experiments.

The adsorption isotherms were obtained at a fixed temperature, adsorbent mass, pH, and agitation parameters. Adsorbate concentration was varied in each medium to study the adsorbate concentration effect. Adsorbate concentrations of 1, 2, 2.5, 3, 3.5, 4, 4.5, 5, 10, and 20 mg/L were used. The pH was adjusted according to the results of the PZC test; values of 6.4 and 5.9 were used for deionized water and water/ethanol (95/5), respectively. Approximately 200 mg of PVDF/clay (30% m/m) spheres were used. Previous tests indicated that such mass of spheres yielded a higher adsorption rate for this clay content. The tests were performed on a shaking table, and the speed was maintained at 150 rpm.

The effect of adsorbent mass during adsorption was also evaluated. In this test, only the amount of adsorbent was varied, and all the other parameters were fixed. The test was conducted using 12 samples with various numbers of spheres (5, 10, 15, 20, 25, 30, 35, 40, 45, 50, 55, or 60). The mass ranged from approximately 26 mg (5 spheres) to approximately 310 mg (60 spheres). The solution volume was held at 25 mL, and the pH was 6.4 for the aqueous medium and 5.9 for the ethanolic medium. The copper concentration of the medium was 5 mg/L for all samples. All tests were performed under stirring at 150 rpm on a shaking table for 24 h. The copper content was determined using atomic absorption spectroscopy.

To investigate the adsorption kinetics, PVDF/clay samples of approximately 200 mg were used. The solution volume was kept at 25 mL, and the test was performed in 100 mL Erlenmeyer flasks. The pH of the medium was 6.4 for the tests performed in water and 5.9 for the tests in the ethanolic medium. The concentration of the medium was maintained at 5 mg/L, and the contact time of the adsorbent with the copper solution was varied. The tests were performed for 12 contact times: 30 min, 1, 2, 3, 4, 5, 6, 7, 8, 12, 18, and 24 h. The tests were performed under stirring at 150 rpm on a shaking table, and each sample was removed after the specified contact time. The copper concentration was determined using atomic absorption spectroscopy.

## 3. Results

Figure 3 presents the X-ray diffraction (XRD) results for the montmorillonite clay, PVDF, and composites. The peaks at 2θ = 20.05° and 36.11° were associated with the β phase of the PVDF, and those at 18.46° and 26.64° are characteristic of the α phase [11,12]. For the clay, diffraction patterns attributed to montmorillonite were observed at 5.72°, 8.96°, and 19.88°. The most intense peak at 26.70° indicates the presence of quartz. Other peaks associated with quartz were observed at 20.98° and 36.67°. The peak at 35.05° can be attributed to aluminum oxide [13]. The XRD patterns of the composites show that the characteristic peaks of PVDF decrease in intensity with increasing clay content. This result may be attributed to the decrease in the PVDF content of the composites [14].

Thermal analysis results are presented in Figure 4. PVDF degradation exhibits a mass loss between 410 °C and 480 °C. During degradation, PVDF releases hydrogen and fluorine, which combine to form hydrogen fluoride, the main product of this process. Thus, the carbon atoms become free to bond with each other, producing the monomer (CH_2_=CF_2_) [15,16,17]. A third residue, C_4_H_3_F_3_, is generated in small quantities (Figure 4a) [15]. The pure clay exhibits a mass loss of approximately 4% up to 120 °C, which is due mainly to water loss.

The composites showed good thermal stability overall up to approximately 200 °C. The composites show an initial mass loss in the range of 300–400 °C due to water loss by the clay. This loss is largest for the composites that have a higher content of montmorillonite, reaching 5% at a content of 30% (m/m). The second stage, in the range of 400–500 °C, corresponded to the mass loss resulting from PVDF degradation, which was initiated earlier in the composites with higher clay content, as shown in Figure 4a. In matrices where a clay-exfoliated structure is present, the degradation temperature generally increases because the clay lamellae act as a barrier to the volatile gases produced by degradation [16]. However, in structures that are not exfoliated, as in this work, the clay acts as a catalyst that decreases the polymer degradation temperature. This effect results from the metal ions present in montmorillonite, which can initiate degradation reactions [17].

The differential thermogravimetry (DTG) curves are illustrated in Figure 4b. This evidences that the first mass loss (300–400 °C) is more intense for the composites with higher clay contents, which is consistent with the higher mass loss due to water loss in these samples. The clay encapsulation by the polymer hinders the release of water vapor from the clay, thus shifting the evaporation peak of water towards higher temperatures. The higher clay content causes the second peak to shift leftward, indicating the catalytic effect of montmorillonite caused by the increase in the degradation rate due to the presence of metallic ions [17].

Figure 5 and Figure 6 show SEM micrographs of the fractured samples. The images provide evidence of the presence of three distinct regions within the material: an internal porous structure, which is overlaid by a denser layer with only a few visible pores, and an outer composite layer composed of radial pores or channels. The composite interior structure consists mainly of interconnected PVDF microspheres and macrovoids. The presence of this structure indicates a longer period of exchange between the solvent and nonsolvent. The reason may be the rapid formation of the thin dense layer on the surface and the hydrophobicity of PVDF, which causes water to enter the sphere more slowly [18,19]. In these cases, there is a certain delay in the liquid–liquid separation, and crystallization can occur to form these spherical structures [20]. A similar dense layer appears in films produced by the immersion precipitation technique [18]. Because PVDF is insoluble in water, it is expected to accelerate liquid–liquid separation, which explains the radial pores layer on the surface of the spheres. According to Figure 6a, the clay insertion in the spheres appears to reduce the thickness of the dense intermediate layer and widen the radial pores in the first layer (Figure 6b). This is a desirable outcome for enhancing the efficiency of absorption processes.

### 3.1. Effect of Adsorbent

Figure 7 shows the effect of clay on the adsorption process. In both media, the degree of absorption increased with increasing clay content. The obtained result was anticipated as clays possess a high surface area and mineral components with a negative charge, making them suitable for metal adsorption by attracting positively charged ions from metal compounds. A total of 200 mg of PVDF/clay_30% spheres removed approximately 32.4% (Figure 7a) and 46.8% (Figure 7b) of the copper ions in an aqueous and ethanolic medium, respectively. The ethanolic medium demonstrated favorability toward the adsorption process, except for the samples containing PVDF and PVDF/clay_5%. The removed concentration was the same in both media (14.7%) for the 5% (m/m) content. These results may indicate that the ethanolic medium more easily reaches the adsorption sites inside the spheres [21]; this suggests that ethanol decreases the surface tension of the solution, which facilitates its dispersion of the liquid inside the sphere.

Figure 8 illustrates the adsorption as a function of composite mass for a clay content of 30%. Figure 8 supports the idea that a higher content of clay in the composite leads to an increased number of active sites available for metal adsorption, as evidenced by the greater adsorption capacity observed with a greater amount of composite. It is also observed that in the ethanolic medium (Figure 8b), a higher copper removal rate was obtained for all the masses used, which is consistent with the hypothesis that ethanol has a greater capacity to wet the surface of the sphere and, therefore, penetrate its internal pores. In the aqueous medium (Figure 8a), the highest removed content of copper ions from the medium (approximately 41%) was obtained at a composite mass of approximately 210 mg, and subsequent additions of spheres did not result in greater removal of the metal from the medium. This result may indicate that saturation of the adsorption sites may already be occurring at this mass, which again may be related to the fact that it is difficult for water to reach them. This phenomenon did not occur in the measurements made in water/ethanol (95/5), where the removal of copper ions continued to increase at composite masses above 210 mg; thus, it was not possible to establish a saturation limit in this test. In this case, the highest removal rate of copper ions, approximately 68%, was obtained when approximately 310 mg (60 spheres) was used.

### 3.2. Adsorption Isotherms

Figure 9 shows the results of applying various models to the aqueous medium containing 5 mg/L of copper. Three isotherm models were tested: Langmuir, Freundlich, and Brunauer–Emmett–Teller (BET). The nonlinear forms of the models were used because the BET model has three degrees of freedom, and thus it cannot easily be linearized and compared with the others [22].

The distribution of the sample data suggests that the data are best described by a sigmoidal function. In addition, the curve obtained for the aqueous medium resembles the Type II curve according to the classification of Brunauer et al. The observed outcome suggests that the adsorption process could involve more than one layer, as the concentration of adsorbate is significant enough to sustain this phenomenon beyond the formation of an initial monolayer [23].

A Type II isotherm is generally associated with nonporous materials or with the presence of pores with dimensions larger than micropores [24], as observed in the micrographs. This finding is in agreement with the SEM images, which show that the outer layers of the material have few pores, mainly the middle layer. This characteristic, in conjunction with the hydrophobicity of PVDF, may hinder the entry of water into the internal sphere. The maximum adsorption capacity (q*_max_*) was 0.361 mg Cu/g.

Figure 8b shows the results for the water/ethanol (95/5) medium. Again, the BET model was found to best describe the data [25,26,27]. However, the Langmuir model has a similar correlation coefficient, which may indicate greater agreement with Type I curves, as classified by Brunauer et al. [25,26,27,28]. However, although the curve for the BET model is not clearly a Type II curve, it does not reach a well-defined plateau of q_e_, which occurs in Type I or Langmuir curves. This finding may indicate that the process does not occur when a monolayer is formed and that the curve could also exhibit Type II behavior at concentrations well above those tested.

It may be more difficult for pure water to penetrate the pores of the spheres, which would limit the availability of adsorption sites for copper ions. This characteristic may cause more rapid saturation of the adsorption sites in the aqueous medium, which may explain the difference in the shapes of the two curves. Another important characteristic that may be related to the hydrophobicity of the material is the maximum adsorption load. When the BET model was applied to the ethanolic medium, the maximum charge was 0.724 mg Cu/g, whereas the value for water was 0.361 mg Cu/g. These data, again, may be related to the hydrophobicity of the material, which hinders the arrival of copper ions at the adsorption sites. It is important to emphasize that these values are still below those for low-cost commercial adsorbents, such as activated carbon and clay [29]. This fact can be explained by the small amount of clay in the material and the characteristics of PVDF, as well as the size of the spheres and the difficulty in overcoming their surface tension. However, this behavior does not disqualify it as a candidate for adsorption applications.

### 3.3. Adsorption Kinetics

In both media, it was observed that considerable time is required for the adsorption reaction to become stable (Figure 10). In water, this time is 15–18 h, whereas, in the ethanolic medium, no plateau was observed, which may indicate that even after 24 h, adsorption was still occurring. This behavior is consistent with the results obtained by the analysis of the isotherms, which indicated that the material has a low adsorption capacity. In both cases, the pseudo-second-order model was found to better describe the experimental data for the composites. For the aqueous medium, a correlation index of 0.976 was observed for this model, compared to a value of 0.930 for the pseudo-first-order model. In the ethanolic medium, the values were 0.984 and 0.962, respectively.

Thus, the second-order kinetic models, considering their assumptions, may suggest that sorption occurs only at localized sites and ions do not interact; the adsorption energy does not depend on the surface coverage; the maximum adsorption capacity corresponds to that of a monolayer of adsorbate on the surface of the adsorbent, and the metal concentration is constant; the metal adsorption follows a second-order equation, as shown in Equation (1).
2S + M = M(S)_2_(1)
where S represents the adsorption sites, and M represents the adsorbate. The pseudo-first- and pseudo-second-order models assume that the slowest step of the reaction is adsorption itself and not the previous step of diffusion in the external medium or pores. Ho and McKay [30] found that the pseudo-second-order model, in particular, describes mainly chemisorption (Figure 11).

### 3.4. Application and Column Tests

Finally, the efficiency of the spheres was evaluated by tests in columns containing the adsorbent. The columns were prepared in 10 mL syringes. Approximately 2 g of the material was packed inside each syringe. In order to avoid loss of material and keep the column well-packed, glass wool was used above and below the column. Three samples of cachaça from two commercial sources were used. Sample A (39% alcohol content) has the brand name Pirassununga 51 and is produced by Companhia Müller de Bebidas in the State of São Paulo. Sample B is the same brand but contains 5 ppm added copper (Cu^2+^). Sample C (40% alcohol content) is an artisanal cassava spirit produced in the State of Maranhão. Figure 12 illustrates the test setup. The syringe (2) was coupled to a 125 mL kitassato (3). A vacuum tube (1) was used to create a vacuum inside the kitassato to ensure a continuous flow of the liquid.

In order to determine the feasibility of applying the spheres for copper removal, adsorption tests of the copper present in commercial cachaça were performed by passing the liquid through adsorption columns containing the compacted adsorbent. Before the tests, the copper contents of the samples were determined by atomic absorption spectroscopy. The results are presented in Table 2.

In sample A, an average copper content of 1.80 mg/L was found. The MAPA Normative Instruction No. 15, March 31 (2011), establishes a concentration limit of 5.00 mg/L. The obtained copper content meets the maximum concentration of 2.00 mg/L allowed by several countries, which is important when cachaça is intended for export [31]. In sample B, which was intentionally contaminated with copper, an average concentration of 6.96 mg/L was observed. Finally, sample C had a copper concentration of 6.56 mg/L, which exceeds the limits set by Brazilian legislation. Because it is a craft cachaça, this drink may have been produced in copper containers. When it is produced in containers made of other materials, such as stainless steel, sulfur components can form, which may greatly reduce the organoleptic quality of the beverage. Copper can catalyze some chemical reactions during distillation, which contributes to the elimination of unpleasant odors during production. However, CuCO_3_Cu(OH)_2_ can form on the internal walls of copper stills, where it can be dissolved by the acidic alcohol vapor and contaminate the distillate, resulting in copper contents exceeding the legal limit [32]. Adsorption tests of the three samples were performed by passing each liquid through a column containing the adsorbent material.

Two tests were performed; in one, the whole spheres were used, and in the other, the spheres were cut in half with the aid of a scalpel. The second test was necessary to evaluate whether the denser intermediated layer that formed during the production of the spheres could affect the adsorption process. In these tests, it was important to use a low flow rate; thus, a vacuum tube was used instead of a vacuum pump, which would cause the residence time of the liquid in the column to be very low. A total of 10 milliliters of each sample was used, and the average flow rate was 0.5 mL/min. The results are shown in Table 3.

For the columns containing the whole spheres, the average removal rates varied between 20 and 25%. For sample A, a final copper concentration of 1.44 mg/L was obtained, which is equivalent to a removal rate of 20.00%. For Sample B, the final concentration was 5.16 mg/L, which represents a removal rate of 25.86%. This sample had the highest initial concentration of copper, and the removal rate was too low for the drink to fall within the standards set by current legislation. For Sample C, the final concentration was 4.96 mg/L, and the average removal rate was 24.39%.

When the spheres were cut, average removal rates between 53 and 56% were observed. These values are more than twice those for the whole spheres. For Sample A, a final concentration of 0.80 mg/L was obtained, which is equivalent to a removed content of 55.56%. For Sample B, the final concentration was 3.20 mg/L, which represents a removal rate of 54.02%. In this case, the drink is within the limits prescribed by current legislation and can be considered safe for consumption.

For Sample C, the final concentration was 3.04 mg/L, and the average removed content was 53.66%. When the cut spheres were used, it was observed during the experiment that the liquid could more easily wet all the material present, whereas, in the experiments in which whole spheres were used, the liquid tended to percolate only along certain paths, which could reduce the efficiency of the material. As the dense layer on the surface of the spheres had less effect, the liquid more easily penetrated the spheres and, thus, reached more adsorption sites, explaining the higher removal efficiency in this case.

Although the copper ions were not completely removed, the use of the material was found to be effective for bringing the three samples within the legislation standards in force in Brazil, where the work was developed. Note that the total removal of copper may not be desirable because it can affect, for example, the sensory qualities of the beverage [33].

## 4. Conclusions

PVDF spheres containing clay particles were obtained by a phase inversion method. Adsorption tests showed that the material is promising for the adsorption of copper ions in both water and an ethanolic medium, where the results depend on clay content. Similar results were obtained in tests where the amount of adsorbent varied.

The X-ray diffractometry data of the samples showed the characteristic peaks of clay throughout the spheres, demonstrating the incorporation of clay. The composites exhibited good thermal stability up to approximately 200 °C. The SEM images showed that the spheres have an internal porous structure with a denser external surface layer, which is attributed to the hydrophobicity of PVDF and apparently affected the batch adsorption process.

The effect of the medium on adsorption was also very clear. In all tests, the ethanolic medium was found to be more efficient, sometimes exhibiting almost twice the adsorption capacity of deionized water under the same conditions.

An analysis of the adsorption isotherms showed that the BET model best describes the experimental data. This result may indicate that adsorption occurs in multiple layers and may also occur on a nonporous surface or a surface with pores with dimensions larger than those of micropores, which was confirmed by the analysis of SEM images.

The application of the spheres in the adsorption of copper ions in cachaças was found to be feasible. Spheres cut transversely showed higher adsorption because the effect of the external layer was reduced, and thus copper ions had greater access to the adsorption sites. For an artisanal cachaça, which had a copper content exceeding that allowed by current legislation, it was possible to remove an average of 54% of the ions, validating the obtainment of samples and their application.

## Figures and Tables

**Figure 1 polymers-15-02643-f001:**
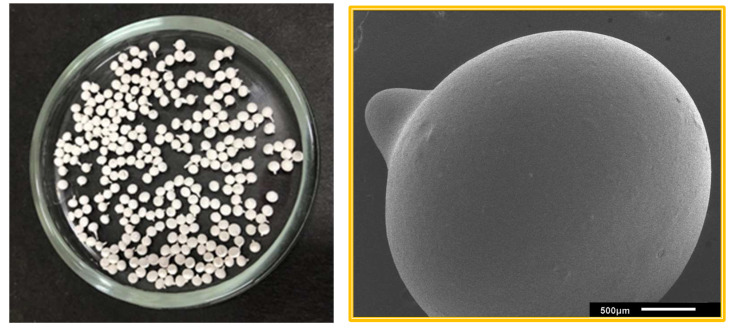
Digital photograph of spheres obtained by the drip method, with a close-up micrograph of a sphere containing 30% incorporated clay.

**Figure 2 polymers-15-02643-f002:**
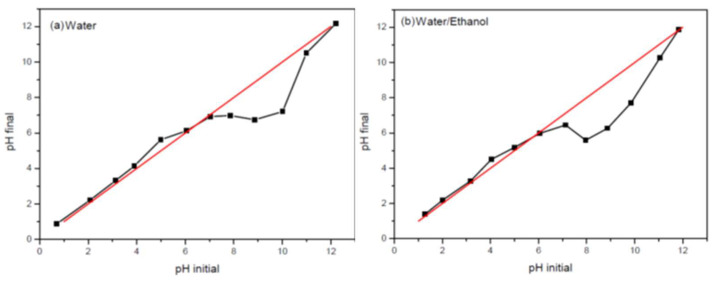
Zero-charge potential (PZC) results for (**a**) aqueous medium and (**b**) water/ethanol medium (95/5 water/ethanol). The PCZ was determined from the intersection of the experimental curve delimited by (initial pH × final pH) (black curve) with the curve in which initial pH = final pH (red curve).

**Figure 3 polymers-15-02643-f003:**
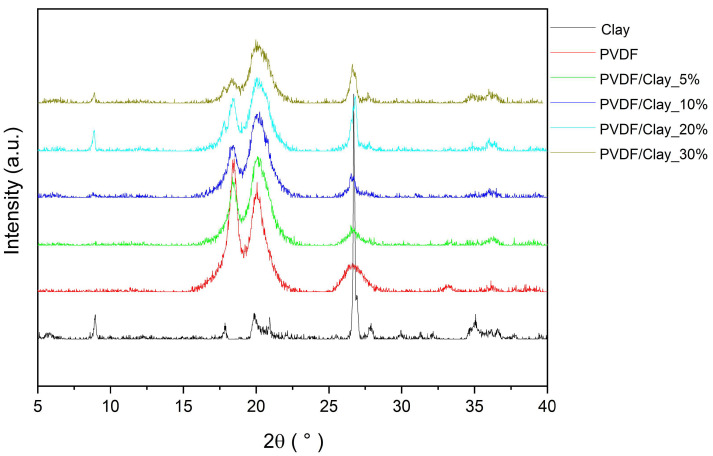
XRD results for montmorillonite clay, PVDF, and PVDF/clay composites.

**Figure 4 polymers-15-02643-f004:**
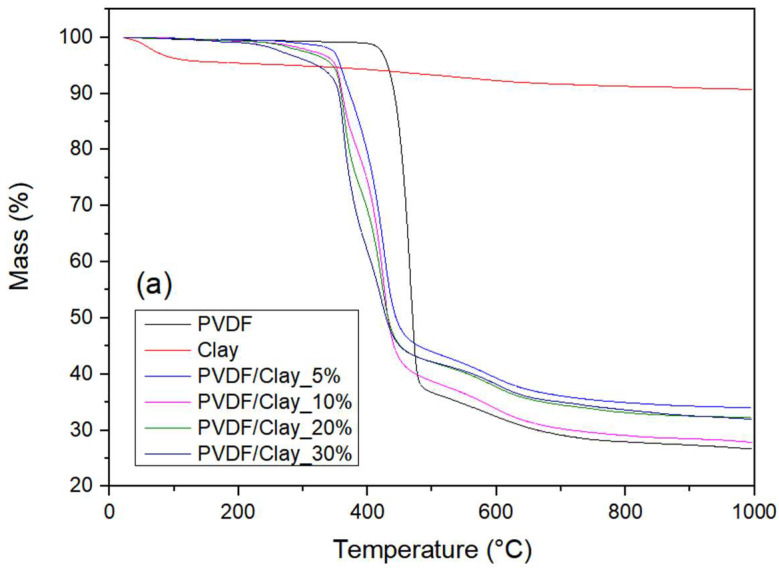
(**a**) TG curves and (**b**) DTG curves of PVDF, clay, and composites.

**Figure 5 polymers-15-02643-f005:**
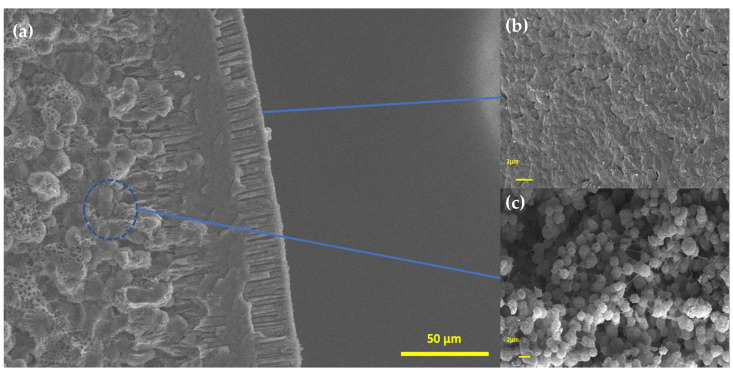
SEM images of fractured PVDF sample: (**a**) cross-section, (**b**) surface, and (**c**) interior.

**Figure 6 polymers-15-02643-f006:**
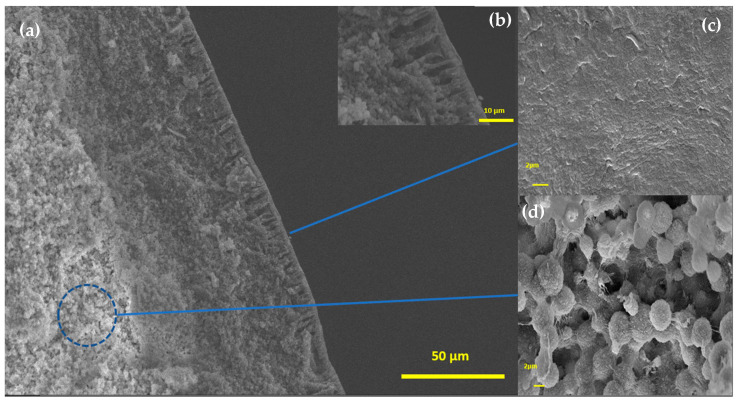
SEM images of PVDF/clay (30% m/m) sample: (**a**) cross-section, (**b**) inset highlighting the pores present in the first layer, (**c**) surface, and (**d**) interior of PVDF/clay spheres.

**Figure 7 polymers-15-02643-f007:**
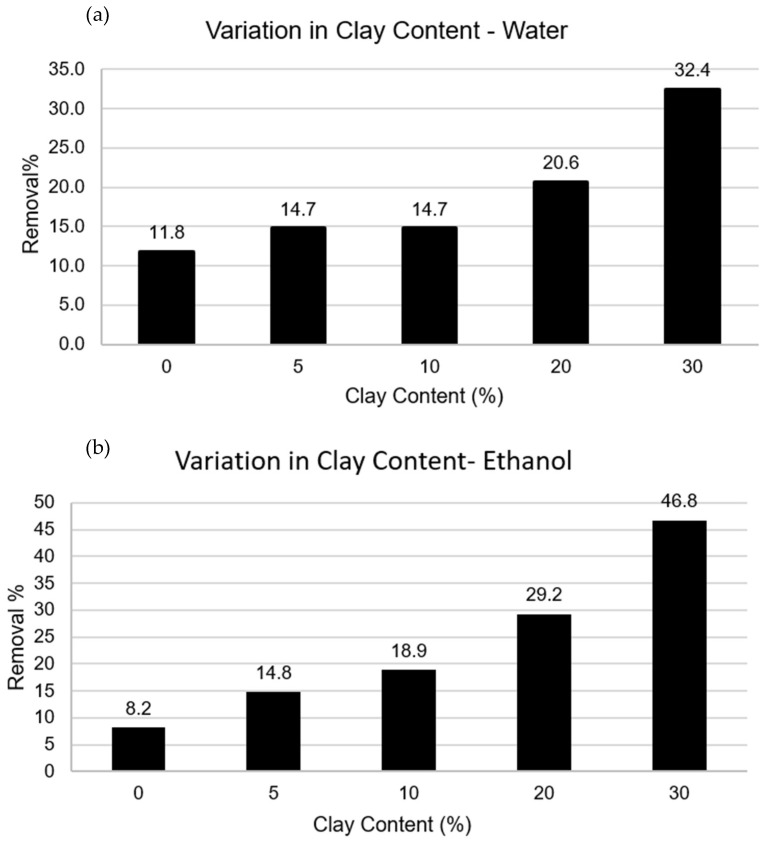
Effect of clay content on copper adsorption in an (**a**) aqueous medium and a (**b**) water/ethanol medium (95/5).

**Figure 8 polymers-15-02643-f008:**
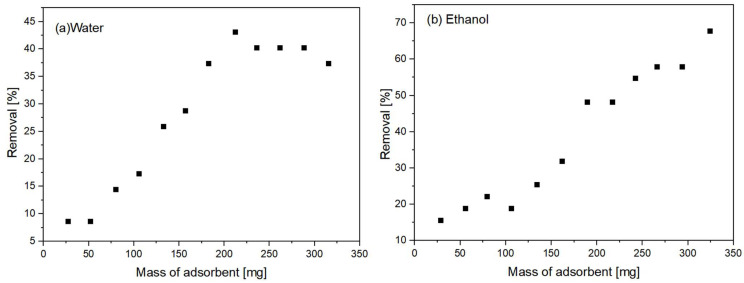
Effect of adsorbent mass on copper adsorption in (**a**) an aqueous medium and (**b**) an ethanolic medium for samples with 30% m/m content.

**Figure 9 polymers-15-02643-f009:**
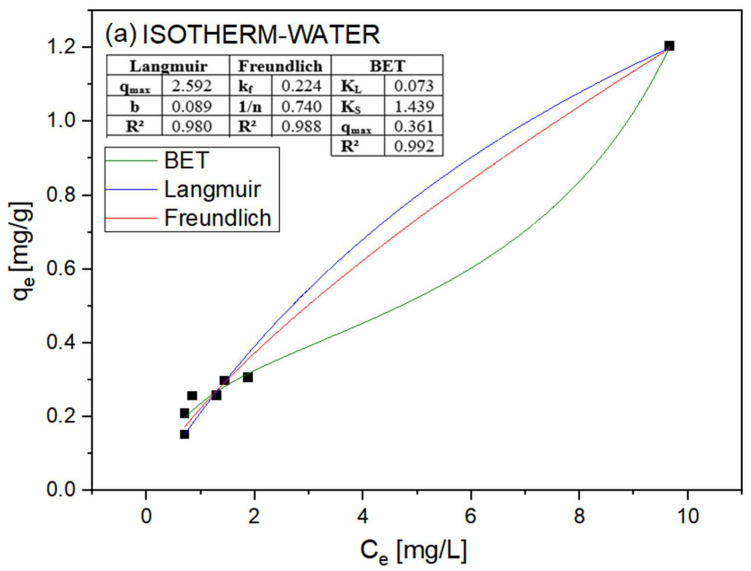
Application of adsorption isotherm models in a (**a**) water/ethanol (95/5) medium and (**b**) in an aqueous medium for PVDF/clay (30% m/m).

**Figure 10 polymers-15-02643-f010:**
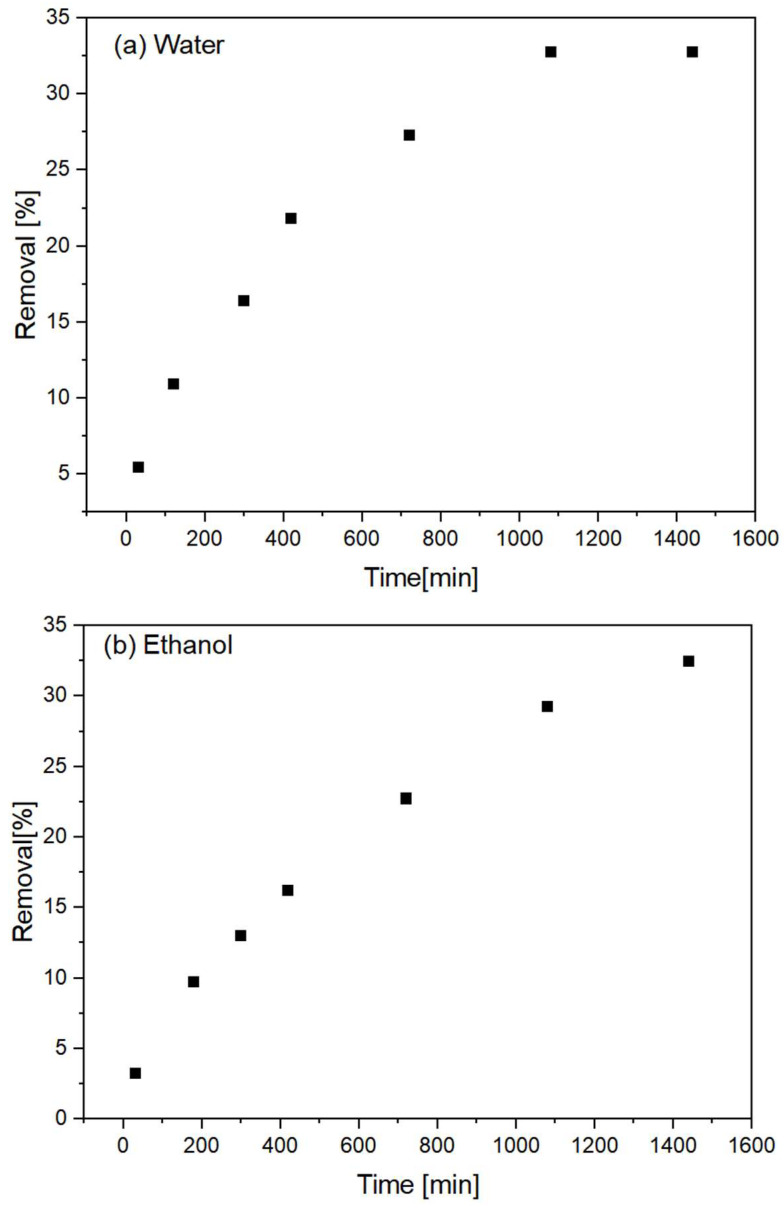
Adsorption kinetics test results for an (**a**) aqueous medium and a (**b**) water/ethanol medium (95/5) for PVDF/clay (30% m/m).

**Figure 11 polymers-15-02643-f011:**
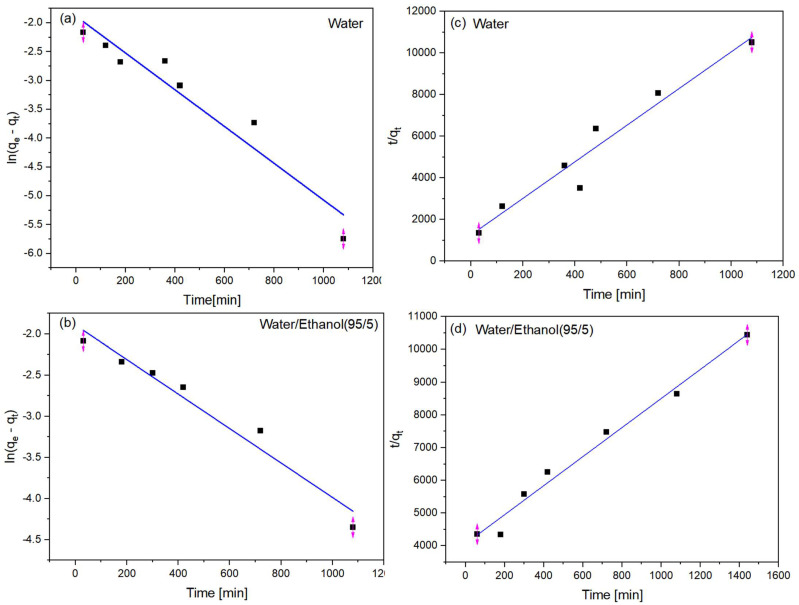
Application of (**a**,**b**) pseudo-first-order; (**c**,**d**): pseudo-second-order kinetic models for aqueous and water/ethanol media (95/5) for PVDF/clay (30% m/m).

**Figure 12 polymers-15-02643-f012:**
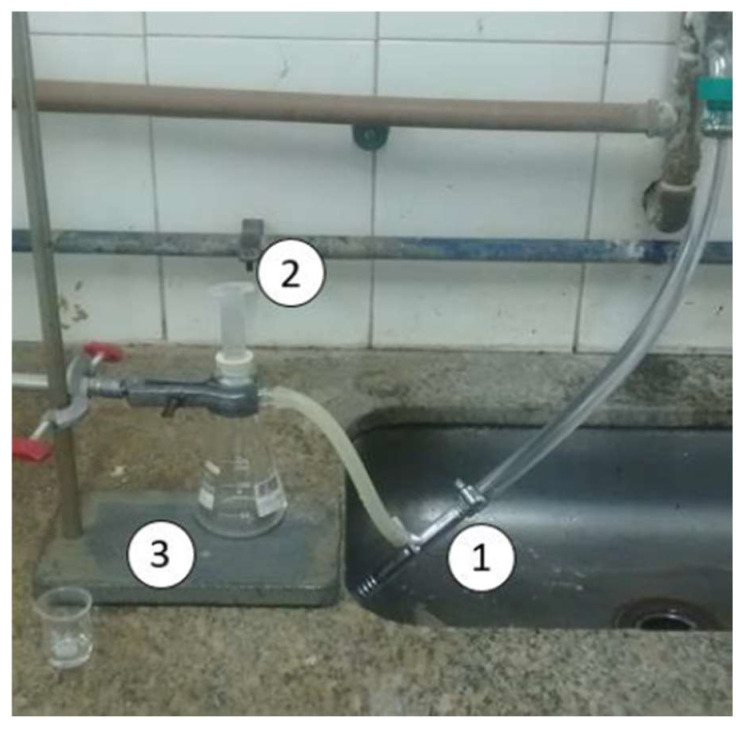
Schematic diagram of column adsorption test.

**Table 1 polymers-15-02643-t001:** Quantifying reagents for producing a 10 mL PVDF/clay mixture.

Clay Content [m/m]	Clay Dispersion		PVDF Solution	
	PVDF [g]	DMF [mL]	PVDF [g]	DMF [mL]
0	-	-	3.0	10.0
5	0.15	2.0	3.0	8.0
10	0.30	2.0	3.0	8.0
20	0.60	2.5	3.0	7.5
30	0.90	2.5	3.0	7.5

**Table 2 polymers-15-02643-t002:** Initial copper concentration of tested samples.

Initial Concentration [mg/L]	
**Sample A**	1.80 (±0.34)
**Sample B**	6.96 (±0.79)
**Sample C**	6.56 (±0.72)

**Table 3 polymers-15-02643-t003:** Column adsorption test results.

Samples	Whole Spheres	
	Final Concentration [mg/L]	Removal [%]
**Sample A**	1.44 (±0.26)	20.00
**Sample B**	5.16 (±0.65)	25.86
**Sample C**	4.96 (±0.56)	24.39
**Samples**	**Cut Spheres ***	
	**Final Concentration** **[mg/L]**	**Removal** **[%]**
**Sample A**	0.80 (±0.14)	55.56
**Sample B**	3.20 (±0.51)	54.02
**Sample C**	3.04 (±0.34)	53.66

* a total of 2 g was used for the adoption tests.

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
