# Peer review of "PVDF/Clay Spheres Obtained through Phase Inversion for Cu Ion Removal"

_polymers, 2023, doi:10.3390/polym15122643_

Round 1

Reviewer 1 Report

1.     Authors reported the use of clay to enhance the Copper removal efficacy of PVDF

2.     In a general view, the authors need to improve the language

3.     The background of the study is weak, the problem statement is not well articulated. The main novelty of this paper is the sphere clay modification for Copper removal. The author is advised to enhance the advantage of utilization of sphere approach. The similar formulation is usually use for membrane , but why you modified it into spheres ?

4.     All the materials used must be comprehensively mentioned, for instance, the ethanol used is not included in section 2.1, and the specifications also such as the molecular weight, percentage of purity etc.

5.     For the sphere size, what is the suitable size? Why you finally used the size as per reported? As the particle sizes can significantly influence the resultant product.

6.     In addition, information on the active functional group present in the clay. The authors need to provide that information

8.     You need to be very clear in titling the subheadings, Line 73.. solution of what??

9.     Under section 3.1,  the removal of efficiency keeps increasing with the clay. What is the optimal clay dosage?

10.  What are the initial properties of the cachaca before copper removal?

11. Figure 5 need to remove raw info in the graphics e.g time, date etc. Put the scale and magnification on the graphics. What is the most significant observation from these figures?

12. Fig.2 -What is ZCP? Explain more .

13. line 335- what is kitassato?

14. Table 3- Explain in methodology how you cut the whole spehere into half? How you set up the experiment for the cut spheres? it was based on the volumes or grams? 

15. Fig 6- Whatis the ethanol content in cacha prior the analysis? Add the characteristics of chacha. 

Author Response

Reply to reviewers

The authors thank the editor for his work and mediation of the manuscript submission process. The authors are also grateful to the reviewers for their valuable comments, which have greatly helped improve the quality of the paper. The manuscript has been revised and improved according to the comments and suggestions made by reviewers. The reply to reviewers is below and was answered point-by-point. Reviewers’ comments were written in black and our reply to comments written in blue.

Reviewer 2 Report

The article reports on the use of spheres composed of poly(vinylidene fluoride) and clay for the adsorption of Cu(II) from aqueous medium and water/ethanol medium (95/5). However, the following key information is missing. This paper can be accepted after some correction and additions:

1. Some sentences are very long and complex for understanding. For example: "This result was expected because clays are suitable for the adsorption of metals owing to their high surface area and the negative charge of their mineral components, which attract the positive ions of metal compounds".

2. In the copper adsorption experiment, what is the experimental pH value? Adsorbents should be added at different pH conditions for copper ion adsorption experiments.

3. Toxic effects of copper ions on environment and the significance of achieving high selective removal for copper ions should be addressed in the introduction of manuscript.

4. There have been many reports about PVDF and MMT-based adsorbents for copper. You should reflect the relative advantages of the adsorbent in the manuscript.

5. I suggest that relevant experiments of competing ions, such as Ca,Mg, K and Na, should be added.

6. Why is XRD analysis put together with thermal analysis in line 164?

7. Whether the solution concentration or syringe height can be adjusted to avoid ball trailing.

8. Please describe Figure 2 in detail.

9. In the analysis of DTG (Fig. 4b) in lines 191-193, the shift of the peak indicates the catalysis of montmorillonite. Please explain carefully what kind of catalysis of montmorillonite has taken place.

10. The SEM images (Fig.5) described in lines 194-195 of this paper show that there is an internal porous structure covered by a small amount of pores. Please show a clearer and more powerful SEM diagram to confirm the existence of the internal porous structure.

11. There is no first-level heading for the third part

Author Response

(The authors gave the same response as above.)

Round 2

Reviewer 2 Report

accept